# Advances in the Mining of Disease Resistance Genes from *Aegilops tauschii* and the Utilization in Wheat

**DOI:** 10.3390/plants12040880

**Published:** 2023-02-15

**Authors:** Hongyun Kou, Zhenbo Zhang, Yu Yang, Changfeng Wei, Lili Xu, Guangqiang Zhang

**Affiliations:** 1State Key Laboratory of Crop Biology, Shandong Agricultural University, Tai’an 271018, China; 2College of Agriculture and Bioengineering, Heze University, Heze 274015, China; 3Shandong Shofine Seed Technology Co., Ltd., Jining 272400, China

**Keywords:** wheat, *Aegilops tauschii*, genetics, genomics, improvement, resistance gene

## Abstract

*Aegilops tauschii* is one of the malignant weeds that affect wheat production and is also the wild species ancestor of the D genome of hexaploid wheat (*Triticum aestivum*, AABBDD). It contains many disease resistance genes that have been lost in the long-term evolution of wheat and is an important genetic resource for the mining and utilization of wheat disease resistance genes. In recent years, the genome sequence of *Aegilops tauschii* has been preliminarily completed, which has laid a good foundation for the further exploration of wheat disease resistance genes in *Aegilops tauschii*. There are many studies on disease resistance genes in *Aegilops tauschii*; in order to provide better help for the disease resistance breeding of wheat, this paper analyzes and reviews the relationship between *Aegilops tauschii* and wheat, the research progress of *Aegilops tauschii*, the discovery of disease resistance genes from *Aegilops tauschii*, and the application of disease resistance genes from *Aegilops tauschii* to modern wheat breeding, providing a reference for the further exploration and utilization of *Aegilops tauschii* in wheat disease resistance breeding.

## 1. Introduction

As one of the major crops, wheat is the main grain for about one third of the world’s population, and its yield is of great significance for alleviating global hunger [1]. According to the FAO, pests and diseases cause 20–40% of global food crop losses and losses of USD 220 billion in agricultural trade every year [2]. In the long-term natural selection and artificial selection, wheat has lost many excellent disease resistance genes [3,4], resulting in a single genetic background of wheat.

*Aegilops tauschii* belongs to the genus *Aegilops* in the Triticeae family of Poaceae [5]. It is an annual weed in wheat fields and is also a relative plant of wheat and the donor species of the D genome during the evolution of common hexaploid wheat [6,7]. Compared with the D genome of wheat, *Aegilops tauschii* has a richer genetic diversity and contains many stress resistance, disease resistance, and insect resistance genes, among many other excellent genes, which are an important breeding resource for wheat breeders to improve wheat traits as well as disease resistance and stress resistance [8,9,10,11]. Therefore, fully exploring and utilizing the excellent disease resistance genes in *Aegilops tauschii* are of great value for wheat disease resistance breeding.

## 2. The Relationship between *Aegilops tauschii* and Wheat

*Aegilops tauschii* (DD, 2n = 14), which originates from West Asia, is mainly distributed in the Middle East, Europe, West Asia, and other places [12,13]. In China, *Aegilops tauschii* is mainly distributed in Xinjiang and the Yellow River Basin (including Shaanxi and Henan Provinces) [14,15]. There are two main lines of introducing *Aegilops tauschii* into the Yellow River Basin of China: first, Middle East→Russia→Xinjiang, China→Yellow River Basin into China; second, it was directly imported from the Middle East via the ancient Silk Road without passing through Xinjiang [16] (Figure 1a,b). As the donor of the D genome of wheat, *Aegilops tauschii* has a wider distribution area and higher genetic diversity than common hexaploid wheat. The addition of the D genome makes hexaploid wheat more adaptable to continental climates, which has laid a solid foundation for the large-scale cultivation of wheat [17].

Wheat originated in the fertile crescent of the Middle East. The first domesticated wild wheat was *Triticum monococcum*, followed by cultivated emmer wheat (*Triticum turgidum*); finally, the common wheat (*Triticum aestivum*) was formed through natural hybridization between cultivated emmer wheat and *Aegilops tauschii* along the Caspian Sea coast [18]. As for how wheat evolved from diploid wheat to the present allohexaploid wheat, there are two theories: one is formation by direct homoploid hybridization. Marcussen et al. found that approximately 6.5 million years ago, the wheat lineage (*Triticum* and *Aegilops*) began to differentiate from a common ancestor into the A and B genome lineages. About 5.5 million years ago, the first hybridization occurred between the A and B genome lineages and led to the origin of the D genome lineage. Furthermore, the second hybridization between the A genome donor *Triticum urartu* (AA) and a related species (BB) of *Aegilops speltodies* occurred approximately 0.8 million years ago, resulting in allotetraploid emmer wheat (*Triticum turgidum*; AABB), which then acclimated to cultivated tetraploid wheat, being crossed again with the D genome donor *Aegilops tauschii* (DD) about 0.4 million years ago to form hexaploid wheat; finally, it was acclimated to *Triticum aestivum* (AABBDD) [19]. In addition, Li et al. re-evaluated the homoploid hybrid origin of *Aegilops tauschii*. Based on the whole chloroplast genome sequence, they analyzed the neighbor joining tree of the *Triticum*–*Aegilops* complex and found that the chloroplast topology reveals that *Aegilops tauschii* is cladistically nested between the A and remaining S * and M genomes. They gave two possible explanations, i.e., the chloroplast capture model and the ancestry capture model. Therefore, they clearly pointed to a more complex history of *Aegilops tauschii* than that proposed by Marcussen et al. [19], one that may have involved multiple rounds of both recent and ancient hybridizations [20]. Then, *Aegilops tauschii* hybridized with tetraploid wheat to form hexaploid wheat. According to Alison et al., wheat was introduced into China in at least two ways. The first is that wheat came from the northwest of West Asia, moving through Eurasia, southern Siberia, and Mongolia to the middle and lower reaches of the Yellow River; the second is that wheat came from West Asia, potentially moving through the Afghanistan or Central Asian oases and northern Xinjiang into China’s Yellow River Basin, rather than the Eurasian steppe [21] (Figure 1c,d). China is the secondary origin center of wheat, mainly including three wheat subspecies: Yunnan wheat (*Triticum aestivum* ssp. yunnanense King), Tibet semi-wild wheat (*Triticum aestivum* ssp. *tibetanum* Shao), and Xinjiang wheat (*Triticum aestivum* ssp. *Petropavlovsk yi*) [22].

Since the domestication of wheat, yield improvement has always been the focus of wheat research. By improving cultivation measures and breeding techniques, the yield per unit area of wheat has been greatly increased [23,24,25,26,27,28,29,30,31,32]. Among them, enhancing wheat disease resistance has always been an important part of improving wheat yield. *Aegilops tauschii* contains abundant beneficial genes for disease resistance, insect resistance, cold resistance, and high quality, which is of great significance for improving wheat yield.

By analyzing the relationship between the two species, it has been proved that *Aegilops tauschii* is an important germplasm resource of wheat. During the long-term evolution process of wheat, the genetic diversity of the D genome has gradually narrowed, resulting in few available genetic resources. However, as the donor of the D genome of wheat, *Aegilops tauschii* carries abundant excellent genes, and the genetic material between them can be exchanged and recombined, which provides a valuable genetic resource pool for wheat breeding. Since both of these wheat species originated in West Asia, we can further enrich wheat germplasm resources by exploring wild wheat and *Aegilops tauschii* from West Asia.

## 3. The Research Progress of *Aegilops tauschii*

At present, the research on *Aegilops tauschii* mainly includes the construction of the *Aegilops tauschii* gene map, the utilization of distant wheat germplasm resources, and the control of malignant weeds in crop fields.

(1) Construction of *Aegilops tauschii* gene map: Jia et al. [33] used AL8/78 as the research material to construct the *Aegilops tauschii* gene map and sequenced the entire genome of AL8/78 with Illumina high-throughput sequencing technology to obtain the whole genome sketch of *Aegilops tauschii*. Luo et al. [34] sequenced the BAC library of AL8/78 using the snapshot method, developed an SNP chip, and constructed a physical map containing 7185 markers, which laid the foundation for the analysis of *Aegilops tauschii*. Luo et al. [35] generated a reference-quality genome sequence for *Aegilops tauschii* strangelata accession AL8/78 by using ordered clone genome sequencing, whole-genome shotgun sequencing, and BioNano optical genome mapping, which is closely related to the wheat D genome. Zhao et al. [36] used new sequencing assembly technology to assemble AL8/78 and obtain new genomic data. Previous studies on the construction of the *Aegilops tauschii* gene map, on the one hand, provided data support for fragment location and cloning of *Aegilops tauschii* itself; on the other hand, they provided technical support for genetic improvement of wheat breeding and made important contributions to the research and utilization of *Aegilops tauschii* resources.

(2) Utilization of distant wheat germplasm resources: Previous studies on the germplasm resources of *Aegilops tauschii* mainly focused on the exploration and utilization of disease resistance genes, insect resistance genes, high-yield genes, etc. The development of disease resistance genes is the focus of wheat germplasm resources. There is a lot of research in this area. The second part of this paper mainly introduces the research in this area in detail. At present, there are eight permanently named cereal cyst nematode resistance genes, namely, *Cre1*–*Cre8*, and only two genes, *Cre3* and *Cre4*, were derived from *Aegilops tauschii* [37,38]. Later, Lage et al. [39] found a gene resistant to wheat aphids on the genome of *Aegilops tauschii*. In the study of high-yield genes, Wan et al. [40] found a major QTL for leaf sheath hairiness (LSH) on *Aegilops tauschii* 4DS, and the allele of this QTL locus was significantly positively correlated with the increase in grain yield, grain weight, and grain weight per spike. Delorean et al. [41] sequenced 273 accessions spanning the known diversity of *Aegilops tauschii*. They found that *Aegilops tauschii* is a reservoir for unique *Glu-D1* alleles and provides a genomic resource for improving wheat quality.

(3) Control of malignant weeds in crop fields: In the current production, the herbicide mesosulfuron–methyl is mainly used to control *Aegilops tauschii* in wheat fields, and it is often used in combination with the safener mefenpyr–diethyl [42]. However, Yuan et al. [43] found that the tolerance of *Aegilops tauschii* to mesosulfuron–methyl was significantly increased in the presence of mefenpyr–diethyl by performing a bioassay, and they proposed that seed dressing with mefenpyr–diethyl could replace spraying to improve the resistance of wheat to mesosulfuron–methyl and enhance the control effect on *Aegilops tauschii*.

## 4. Discovery of Disease Resistance Genes from *Aegilops tauschii*

As an important resource of wheat resistance genes, *Aegilops tauschii* provides stripe rust resistance genes, leaf rust resistance genes, powdery mildew resistance genes, brown spot resistance genes, etc.

### 4.1. Discovery of Rust Resistance Genes from Aegilops tauschii

Rust is one of the main diseases of wheat, including three types: stripe rust, leaf rust, and stem rust [44].

#### 4.1.1. Discovery of Stripe Rust Resistance Genes from *Aegilops tauschii*

Wheat stripe rust is a common disease of wheat, caused by *Puccinia striiformis* f. sp. *tritici* (*Pst*), which is characterized by a high prevalence and frequency, a wide incidence range, and serious damage [45]. It is estimated that the annual loss of wheat production caused by stripe rust worldwide is over 5 million tons, with an estimated market value of USD 1 billion [46]. On the one hand, the pathogen of stripe rust invades wheat and absorbs wheat nutrients and water, affecting plant growth; on the other hand, it causes a reduction in the wheat leaf area, affecting photosynthesis, and reducing wheat yield. Using wheat stripe rust resistance genes to control wheat stripe rust is the most effective and environmentally friendly way to counter this disease [47]. At present, there are 84 permanently named stripe rust resistance genes in wheat, namely, *Yr1*–*Yr84*, and only 8 genes, *Yr5*, *Yr7*, *Yr15*, *Yr18*, *Yr27*, *Yr28*, *Yr36*, and *Yr46*, have been cloned [17,48,49,50,51,52,53,54]. Among them, only the *Yr28* gene is derived from *Aegilops tauschii* (Table 1).

*Yr28* was first discovered and named by Singh et al. [55], and it was located on the short arm of the 4D chromosome and had resistance to multiple stripe rust races, showing all-stage resistance (ASR) in *Aegilops tauschii*. Liu et al. [56] and Huang et al. [57] found the dominant stripe rust resistance gene *YrAS2388* in *Aegilops tauschii* and located it on chromosome 4DS. In 2013, Liu et al. [58] found the existence of *YrAS2388* in the subspecies *Aegilops tauschii subsp. Strangulata* near the Caspian Sea. In 2019, Zhang et al. [17] cloned the *YrAs2388* gene using traditional map-based cloning technology, confirmed that the *YrAs2388* gene is the internationally named *Yr28* gene, and introduced this gene into hexaploid wheat using synthetic wheat. *Yr28* encodes a typical NBS-LRR structural protein (*NLR_4DS-1_*). Compared with the susceptible haplotype of *Yr28*, the resistant haplotype has two repeated 3′ untranslated regions (3′UTR1 and 3′UTR2), and there are five transcript variants in the domain of the gene (two alternative splicing variants are associated with 3’UTR1, and the other three alternative splicing variants are associated with 3’UTR2), which makes *Aegilops tauschii* and synthetic wheat containing the *Yr28* gene resistant to stripe rust, but *Yr28* only shows adult plant resistance (APR) in synthetic wheat. Athiyannan et al. [59] found a full-growth period resistance gene, *YrAet672*, from *Aegilops tauschii* CPI110672 and successfully cloned it through map-based cloning. It was proved that the gene was identical to the coding region sequence of *YrAS2388*, only being different in the 5’UTR and 3’UTR regions, and was an allele of *YrAS2388* and *Yr28*. The study also found that the hexaploid wheat genome can inhibit the expression of *YrAet672*, where there may be some modification or inhibition of the gene.

#### 4.1.2. Discovery of Leaf Rust Resistance Gene from *Aegilops tauschii*

Wheat leaf rust is a fungal disease caused by *Puccinia triticina* (*Pt*), which has the characteristics of multiple infection and strong harmfulness [60]. When wheat is infected with leaf rust, the yield decreases by 5–15%, and the yield loss even reaches 40% in pandemic years [61]. Wheat leaf rust fungus mainly affects the normal growth and development of wheat by infecting wheat leaves, especially photosynthesis, and then affects grain filling, resulting in a reduction in the 1000-grain weight [62]. Up to now, 82 wheat leaf rust resistance genes have been identified, namely, *Lr1*–*Lr82* [63,64], although only seven genes, *Lr1*, *Lr10*, *Lr13*, *Lr21*, *Lr22a*, *Lr34*, *Lr42*, and *Lr67*, have been cloned [65,66]. At present, the leaf rust resistance genes from *Aegilops tauschii* include *Lr21*, *Lr22a*, *Lr32*, *Lr39*, and *Lr42* [67,68,69,70,71] (Table 1).

The *Lr21* gene was found in synthetic wheat RL5406 by Rowland and Kerber in 1974 and is located on chromosome 1DS. The gene was derived from *Aegilops tauschii* on the coast of the Caspian Sea [67] and is an all-stage resistance gene. In 2003, Li and Gill found that RGA-like could be used to mark all known members of the *Lr21* leaf rust resistance gene family in *Aegilops tauschii* and wheat, and the *Lr21* gene was successfully cloned using the diploid/polyploid shuttle localization strategy [72,73]. Scofield et al. [74] analyzed the resistance mechanism of *Lr21* using virus-induced gene silencing and found that *Lr21* encodes a leucine-rich repeat resistance gene product at the nucleotide binding site, which may contribute to wheat resistance. To further elucidate the origin of the *Lr21* gene, Huang et al. [75] identified and analyzed three basic non-functional *Lr21* haplotypes, H1, H2, and H3, by analyzing the *Lr21* and *Lr21* allele sequences of 24 wheat cultivars and 25 *Aegilops tauschii* and found that *Lr21* is a chimera of H1 and H2 in wheat. The next year, Fu et al. [76] re-sequenced the wheat leaf rust resistance locus *Lr21* of 95 wheat varieties released in Canada, revealed 13 SNPs, 4 insertions and deletions, 10 haplotypes, and 4 major haplotype groups, and developed a new SCAR marker to identify resistant haplotypes and haplotype groups. In North America, Kolmer and Anderson found that the physiological races TFBJQ and TFBGQ were toxic to wheat varieties containing *Lr21* [77]. Kumari et al. [78] developed a KASPar marker for the *Lr21* gene and tested it on 384 American wheat lines, finding that the marker could effectively distinguish resistant and susceptible genotypes and could be applied to molecular-marker-assisted breeding of disease-resistant wheat varieties through gene pyramiding. Naz et al. [79] studied the evolution and functional differentiation of *Lr21* in diploid and hexaploid wheat by using population genetics and high-resolution comparative genomics and found that there were at least two independent polyploidization events in wheat evolution. At the same time, a unique *Lr21-tbk* allele and its neofunctionalization were discovered in hexaploid wheat, and the seedling resistance and adult plant resistance were related to the development-dependent variation in *Lr21* expression, which helps us to further understand the evolution of *Lr21* and its role in broad-spectrum resistance to leaf rust in wheat.

*Lr22a* was discovered by Dyck and Kerber in synthetic wheat derived from common wheat and *Aegilops tauschii* and mapped on chromosome 2DS [68]. Pretorius found that in adult-plant-resistant wheat line RL6044, *Lr22a* was not expressed at the seedling stage but at the adult stage, indicating that *Lr22a* endowed this line with adult plant resistance (APR) [80]. The next year, Pretorius found that *Lr22a* was a partially recessive monogenic inheritance [81]. In order to select varieties containing the *Lr22a* gene among different wheat lines, Hiebert et al. [82] found that a GWM marker is close to *Lr22a* and could be used as a microsatellite marker of the *Lr22a* gene, and it is useful under different genetic background conditions. Based on TACCA (targeted chromosome-based cloning via long-range assembly), Thind et al. [83] cloned the broad-spectrum leaf rust resistance gene *Lr22a* using molecular marker information and ethyl methane sulfonate (EMS) mutants and found that *Lr22a* encodes an intracellular immune receptor homologous to the RPM1 protein of *Arabidopsis thaliana*. Although *Lr22a* has broad-spectrum resistance and has been successfully cloned, it has not been widely used in production. Sharma et al. [84] identified and isolated SNPs using the *Lr22a* coding sequence and developed four competitive allele-specific polymerase chain reaction (KASP) markers, which can reliably detect the presence or absence of *Lr22a* and will contribute to the application of *Lr22a* in breeding.

*Lr32* is a whole-growth period resistance gene, which was first discovered by Kerber et al. and later located on the 3DS chromosome. Thomas found that the *Lr32* gene has two simple sequence repeat (SSR) loci, wmc43 and barc135, which can be used as superposition markers between *Lr32* and other widely effective leaf rust resistance genes [69,85,86].

*Lr39* was first discovered by Pretorius and located on 2DS by Raupp through microsatellite marker analysis, and it is a full-growth stage disease resistance gene [70,87]. Li et al. [88] revealed 36 differentially expressed genes (DEGs) for wheat leaf rust resistance mediated by *Lr39*/*41* through suppression subtractive hybridization and microarray analysis and quantitatively analyzed the expression levels of eight selected DEGs at different stages of *Lr39/41*-mediated resistance.

*Lr42* is a partially dominant gene, which was discovered and reported by Cox et al. together with the *Lr41* and *Lr43* genes; Sun et al. located it on chromosome 1DS [71,89]. Harsimardeep et al. [90] found that *Lr42* was dominant in *Aegilops tauschii*, fine-mapped the gene to the 3.16 Mb genomic region on chromosome 1DS of Chinese Spring and the 3.5 Mb genomic region on chromosome 1 of the *Aegilops tauschii* reference genome, and developed two co-dominant allele-specific polymorphism (KASP) markers (*SNP113325* and *TC387992*) on the flanking region of *Lr42* for assisted breeding selection. Liu et al. [91] identified the sequence polymorphism of the differentially expressed gene (*TaRPM1*) encoding the hypothetical NB-ARC protein in the *Lr42* candidate region through RNA sequencing of the *Lr42* allelic variation near-isogenic line and developed a diagnostic DNA marker for *Lr42*. The marker is designed based on deletion mutations and single-nucleotide polymorphisms (SNPs) in the gene and has the advantages of a low cost and easy determination. In 2022, Lin et al. identified three candidate genes of *Lr42* using the batch-isolated RNA-Seq (BSR-Seq) mapping strategy. Among them, the gene AET1Gv20040300 has obvious sequence differences in disease-resistant and susceptible varieties. The down-regulation of the *Lr42* gene caused by virus-induced gene silencing (VIGS) and the mutation of the *Lr42* C700Y amino acid caused by mutagenesis were carried out on this gene. It was found that both caused the loss of resistance of the *Aegilops tauschii* line TA2450, and the candidate gene *AET1Gv20040300* was finally determined as *Lr42* and successfully cloned [92].

#### 4.1.3. Discovery of Stem Rust Resistance Genes from *Aegilops tauschii*

Wheat stem rust, caused by *Puccinia graminis* Pers. f. sp. *tritici*, is one of the most devastating fungal diseases in wheat production. The disease can lead to wheat production reductions of up to 75%, and some areas even experience no production [93]. Stem rust mainly destroys the tissues of wheat stems and leaves. Its spores can penetrate the leaves, invade the host from the stomata, reduce the photosynthetic area of the host, destroy the guiding tissues of the stems, and hinder nutrient transport [94]. At present, 62 wheat stem rust resistance genes have been officially named, namely, *Sr1*–*Sr62* [95,96,97,98], of which *Sr13*, *Sr21*, *Sr22*, *Sr33*, *Sr35*, *Sr45*, *Sr46*, *Sr50*, and *Sr62* have been cloned [98,99,100,101,102,103,104,105]. At present, there are three genes, *Sr33*, *Sr45*, and *Sr46*, found to be resistant to stem rust in *Aegilops tauschii* [106,107,108] (Table 1). Furthermore, some stem rust resistance genes are still under investigation and not formally named, such as the *SrTA10187* and *SrTA10171* genes located on chromosomes 6DS and 7DS, respectively [109]. Wiersma et al. finely mapped *SrTA10187* to the 1.1cM region and developed PCR-based SNP and STS markers using genotyping-by-sequencing tags and SNP sequences available in online databases [110].

*Sr33* is an adult plant resistance gene; Kerber and Dyck first discovered it, and then Jones et al. located it on the 1DS chromosome arm of wheat through the double-terminal and normal chromosome 1D recombination substitution line. This gene is derived from *Aegilops tauschii* [111,112]. Han et al. discovered the co-dominant markers *Xbarc152* and *Xcfd15*, located on both sides of *Sr33* [113]. Sambasivam successfully cloned *Sr33* and found that it encodes a coiled-coil, nucleotide-binding, leucine-rich repeat protein, which is closely related to its ability to confer stem rust resistance in wheat [100]. Through bioinformatics analysis, Ivaschuk et al. found that sequences S5DMA6 and E9P785 were the closest homologues of the *Sr33* gene product RGA1e protein [114]. Md Hatta et al. found that *Sr33* functions not only in wheat but also in barley to resist stem rust [115].

*Sr45* comes from *Aegilops tauschii*; it was discovered by Marais et al. and is closely linked to *Sr33* and localized on chromosome 1DS [116]. It is an adult plant disease resistance gene [117]. Therefore, to develop a marker for the identification of *Sr45* in the tight linkage of centromere-*Sr45*-*Sr33*-*Lr21*-telomere, Periyannan et al. fine-mapped the *Sr45* region in a large mapping population generated by the hybridization of CS1D5406 (the disomic substitution line on chromosome 1D of RL5406 replaced Chinese Spring 1D) with Chinese Spring and amplified a fragment linked to *Sr45* using an AFLP marker sequence to mark *Sr45*-carrying haplotypes [107]. Steuernagel used MutRenSeq technology to clone the stem rust resistance gene *Sr45* by combining chemical mutagenesis with exon capture and sequencing [102]. Md Hatta et al. found that *Sr45*, like *Sr33*, could confer stem rust resistance in both wheat and barley [115].

*Sr46* was discovered by Evans but not published in relevant papers; however, it was then included in the “Catalogue of Gene Symbols for Wheat” by McIntosh [108,117]. Yu et al. located *Sr46* on 2DS. The gene is significantly affected by temperature and is an adult plant resistance gene. In the meantime, Yu et al. found that two closely linked markers, *Xgwm210* and *Xwmc111*, could be used for marker-assisted selection of *Sr46* in wheat breeding [108]. Arora et al. combined association genetics with R gene enrichment sequencing (AgRenSeq) to successfully clone the stem rust resistance gene *Sr46* [105]. *Aegilops tauschii* germplasms RL5271 and CPI110672 were resistant to wheat stem rust. Athiyannan et al. identified RL5271 and found that *SrRL5271* was the dominant resistance gene in RL5271, while CPI110672 resistance was separated in *Sr672.1* and *Sr672.2*. They also found that *SrRL5271* and *Sr672.1* have the same sequence and are the alleles of *Sr46*, except that an amino acid sequence (N763K) is different from *Sr46*, although the other amino acid sequences are identical [118].

### 4.2. Discovery of Powdery Mildew Resistance Genes from Aegilops tauschii

Wheat powdery mildew is a widespread wheat disease in the world, which is caused by *Blumeria graminis* f. sp. *Tritici* (*Bgt*). The yield reduction caused by powdery mildew accounts for about 5% of the yield reduction caused by wheat pests and diseases, which seriously affects the yield improvement and quality improvement of wheat [119]. Wheat powdery mildew is a type of living parasitic fungus. When conidia make contact with living tissues such as the leaves and stems of wheat, they will be immersed in host cells to form white flocculent small mildew spots, which weakens plant photosynthesis and enhances transpiration and respiration, resulting in reduced plant dry matter accumulation and reduced yield [120]. Thus far, a total of 68 wheat powdery mildew resistance genes have been officially named, namely, *Pm1*–*Pm68*, and *Pm1a*, *Pm2*, *Pm3b*, *Pm4*, *Pm5e*, *Pm8*, *Pm17*, *Pm21*, *Pm24*, *Pm38*, *Pm41*, *Pm46*, and *Pm60* have been cloned [49,50,121,122,123,124,125,126,127,128,129,130,131]. Meanwhile, researchers have found 140 QTLs for powdery mildew resistance, distributed in 21 chromosomes of wheat, among which 4 QTLs have been confirmed and widely used in some regions or units [132]. Thus far, the powdery mildew resistance genes discovered and officially named from *Aegilops tauschii* include *Pm2a*, *Pm19*, *Pm34*, *Pm35*, and *Pm58* [133,134,135,136,137] (Table 1).

*Pm2a*, a powdery mildew resistance gene, was discovered by Pugsley and Carter in 1953 and later officially named *Pm2*, with whole-growth resistance [133,138]. In 1970, researchers found that *Pm2a* was located near the centromere of wheat chromosome 5DS [139]. Lutz et al. [140] obtained 40 materials containing the *Pm2* gene from 400 *Aegilops tauschii* materials. Sáchez-Martín et al. [123] cloned the wheat powdery mildew resistance gene *Pm2a* using the MutChromSeq (mutant chromosome sequencing) strategy. Since powdery mildew and disease resistance genes are consistent with the gene-for-gene hypothesis, there are corresponding avirulence genes in the pathogen, which react with disease resistance genes to stimulate a wheat disease resistance response [141]. Praz et al. [142] cloned the avirulence gene *BgtE-5845* corresponding to *Pm2* by combining genetic mapping and association analysis, namely *AvrPm2*, and speculated that *AvrPm2* may have dual functions: first, it has the function of recognizing and stimulating the host immune response with *Pm2* in incompatible interactions; second, it participates in the formation of haustoria in the affinity interaction and inhibits the function of the host cell defense response. Manser et al. [143] further studied *AvrPm2* and found two other haplotypes of *AvrPm2*, *AVRPM2-H1* and *AVRPM2-H2*, in powdery mildew strains USA7 and USA2, and only *AVRPM2-H1* could be specifically recognized by *Pm2a*.

*Pm19* is a new powdery mildew resistance gene discovered by Lutz et al. [134] in their progeny by crossing two powdery mildew resistance wheat lines with susceptible durum wheat, and it is located on chromosome 7D.

*Pm34* was discovered by Miranda et al. [135] using the F (2) derivative line of NC97BGTD7×Saluda, and it is a new wheat powdery mildew resistance gene; the authors then marked the gene on the long arm of chromosome 5D with microsatellite markers and officially named it *Pm34*.

*Pm35* is a single gene controlling powdery mildew resistance identified by Miranda et al. [136] through genetic analysis of the F (2) derivative line of NCD3×Saluda. Miranda then located the gene on chromosome 5DL with microsatellite markers, and the gene was independent from *Pm34*, being officially named *Pm35*.

*Pm58* was derived from *Aegilops tauschii* TA1662 near the Caspian Sea. Wiersma et al. used 96 BC_2_F_4_ introgression lines to position *Pm58* within an interval of 8.6 Mb on chromosome 2DS and obtained two high-generation lines carrying the *Pm58* gene and excellent agronomic traits the following year [137,144]. These two lines are highly resistant to powdery mildew, but the yield is lower than that of common wheat. In 2022, Xue et al. [145] fine-mapped the *Pm58* gene into a 141.3 Kb *Xsts20220*-*Xkasp61553* region and developed a co-segregated KASP marker, *Xkasp68500*, that could be used for *Pm58*-assisted selection breeding.

### 4.3. Discovery of Other Disease Resistance Genes from Aegilops tauschii

Besides stripe rust resistance genes, leaf rust resistance genes, and powdery mildew resistance genes, *Aegilops tauschii* also contains septoria tritici blotch resistance genes and brown spot resistance genes. *Stb5* is a septoria tritici blotch resistance gene, which was discovered by Arraiano et al. in Synthetic 6x (derived from a hybrid of *Triticum dicoccoides* and *Triticum tauschii*) and located on the 7D short arm, endowing the plant with resistance at the whole-growth stage [146,147] (Table 1). *Tsr3*, a brown spot resistance gene, from tetraploid wheat and *Aegilops tauschii* synthetic wheat lines (CS/XX41, CS/XX45, and CS/XX110) was identified by Tadesse et al. (Table 1). The *Tsr3* gene is a recessive gene. Tadesse et al. used SSR markers to carry out linkage analysis and found that *Tsn3a* of XX41, *Tsn3b* of XX45, and *Tsn3c* of XX110 were clustered near *Xgwm2a*, located on the short arm of chromosome 3D [148].

**Table 1 plants-12-00880-t001:** Officially named disease resistance genes in *Aegilops tauschii*.

Classification	Gene	Types	Chromosome	Cloned	Reference
stripe rust resistance genes	*Yr28*	ASR	4DS	Yes	[55]
leaf rust resistance genes	*Lr21*	ASR	1DS	Yes	[67]
	*Lr22a*	APR	2DS	Yes	[68]
	*Lr32*	ASR	3DS	No	[69]
	*Lr39*	ASR	2DS	No	[70]
	*Lr42*	ASR	1DS	Yes	[71]
stem rust resistance genes	*Sr33*	ASR	1DS	Yes	[111]
	*Sr45*	APR	1DS	Yes	[116]
	*Sr46*	APR	2DS	Yes	[108]
powdery mildew resistance genes	*Pm2a*	ASR	5DS	Yes	[133]
	*Pm19*	ND	7D	No	[134]
	*Pm34*	ND	5DL	No	[135]
	*Pm35*	ND	5DL	No	[136]
	*Pm58*	ND	2DS	No	[137]
septoria tritici blotch resistance genes	*Stb5*	ASR	7DS	No	[146]
brown spot resistance genes	*Tsr3*	ND	3D	No	[148]

ND: not detected; ASR: all-stage resistance; APR: adult plant resistance.

## 5. Application of Disease-Resistant Genes from *Aegilops tauschii* in Wheat Breeding

The utilization of disease resistance genes in *Aegilops tauschii* is of great significance for expanding wheat disease resistance. Synthetic hexaploid wheat (SHW) is an artificially created hexaploid wheat that can simultaneously introduce genetic variations from tetraploid wheat and *Aegilops tauschii*, and it has been widely used to expand the genetic diversity of common wheat [149]. The method mainly includes two main steps: First, a hybrid F_1_ with an ABD genome is produced by direct hybridization of tetraploid wheat with *Aegilops tauschii*, and then a synthetic hexaploid wheat with an AABBDD genome is obtained through chromosome doubling [149]. Second, the genetic variation in *Aegilops tauschii* and tetraploid wheat is introduced into common wheat varieties by using the artificial synthetic hexaploid wheat as a bridge and common wheat as a backcross or topcross [107].

### 5.1. Application of Rust Resistance Genes from Aegilops tauschii in Wheat Breeding

#### 5.1.1. Application of Stripe Rust Resistance Genes from *Aegilops tauschii* in Wheat Breeding

*Yr28* is the first stripe rust resistance gene cloned from *Aegilops tauschii*. Through map-based cloning results, previous researchers designed resistance co-segregation molecular markers, conducted auxiliary selection, and bred a new variety, Shumai 1675 [17]. The main cultivation processes were as follows: (1) introducing disease resistance genes into synthetic hexaploid wheat; (2) establishing a breeding population; (3) F_2_ small group mixed selection; (4) F_3_ small population for molecular marker selection to prevent target gene loss; (5) F_5_ line focused on the selection of yield-related traits, and molecular marker selection of disease resistance genes [150]. After the above processes, the yield of F_5_ and its selected line Shumai 1675 increased significantly and showed stripe rust resistance.

#### 5.1.2. Application of Leaf Rust Resistance Genes from *Aegilops tauschii* in Wheat Breeding

*Lr21* is the first powdery mildew resistance gene found and successfully cloned from *Aegilops tauschii*. Thus far, the application of the *Lr21* gene in wheat breeding is low. Mebrate et al. [151] used 31 Pt races to detect 36 wheat cultivars from Ethiopia and Germany and found that Sirbo and Granny contained *Lr21*. Gebrewahid et al. [152] identified 83 wheat varieties and 36 lines with known leaf rust resistance (Lr) genes from three provinces in China. There were 41 cultivars containing leaf rust resistance (Lr) genes, but only Wanmai 47 contained *Lr21*. Khakimova et al. [153] studied 36 synthetic hexaploid wheat varieties from Russia and identified 11 materials containing *Lr21*. Zhang et al. [154] identified and analyzed 46 Chinese landraces and found that only Baiheshang contained *Lr21*.

*Lr22a* has broad-spectrum resistance to wheat leaf rust, but it has not been widely used in production due to differences in varieties from different regions. Khakimova et al. [153] studied 36 synthetic hexaploid wheat varieties from Russia and found that three of them contained *Lr22a*. Huang et al. identified and analyzed 36 wheat production varieties in Gansu Province and found that the varieties Huining 15, Lantian 37, and Longjian 113 showed resistance to all *Lr22a* non-toxic races, indicating that these three materials may contain *Lr22a* [155]. However, Atia et al. [156] identified and analyzed 50 wheat varieties in Egypt and successfully identified 21 Lr genes, and all wheat varieties contained *Lr22a*.

Although *Lr32* has not been cloned successfully, it was found that the disease-resistant wheat varieties contained this gene in actual production identification. Zhao et al. identified 23 Chinese wheat microcore collections, and the five core germplasms of Tongjiaba wheat, Honghua wheat, Kefeng 3, Atlas66, and Golden wheat contained the *Lr32* gene [157]. Hanaa et al. [158] identified leaf rust resistance in 10 Egyptian spring wheat varieties at the seedling stage and found that Sids12 and Sakha93 contained *Lr32*. Bahar et al. [159] used SSR markers of 13 resistance genes to identify 57 wheat lines and found that all 57 lines contained *Lr32*. Atia et al. [156] identified and analyzed 50 wheat varieties in Egypt and successfully identified 21 Lr genes, and all wheat varieties contained *Lr32*.

*Lr39* is a powdery mildew resistance gene of wheat at the seedling stage, and it has certain development value [160]. Hanaa et al. [158] identified leaf rust resistance in 10 Egyptian spring wheat varieties at the seedling stage and found that Miser1 and Miser2 contained *Lr39*. Atia et al. [156] identified and analyzed 50 wheat varieties in Egypt and successfully identified 21 Lr genes, and 42 wheat varieties contained *Lr39*. Wang et al. identified the leaf rust resistance of 71 important wheat production varieties in Henan Province and found that four cultivars contained *Lr39* [161].

As early as 1991, *Lr42* was transferred from *Aegilops tauschii* to common wheat through hybridization by the Wheat Germplasm Resources Center (WGRC) of Kansas State University in the United States, and the KS91WGRC11 wheat line was developed [162]. Subsequently, the International Maize and Wheat Improvement Center (CIMMYT) widely applied the disease resistance genes in this line to breeding materials. Through the identification and analysis of 103 wheat varieties (lines) of CIMMYT and 35 control varieties containing known leaf rust resistance genes, Han et al. found that 11 CIMMYT wheat varieties may contain *Lr42* [163]. Liu et al. tested 66 wheat varieties approved by Qinghai Province and found that 23 varieties contained *Lr42*, accounting for 34.85% [164]. Among 52,943 CIMMYT lines or varieties sequenced by GBS, 5121 pedigrees contained *Lr42* [92].

#### 5.1.3. Application of Stem Rust Resistance Genes from *Aegilops tauschii* in Wheat Breeding

*Sr33* is an important gene for resistance to the physiological race Ug99 of stem rust, and its wide application is of great significance to reduce the harm of stem rust. Ma et al. conducted SSR detection on 58 spring wheat varieties resistant to Ug99 introduced at home and abroad and 18 main wheat varieties in Heilongjiang Province and found that only one spring wheat variety material resistant to Ug99 and three main wheat varieties in Heilongjiang Province contained the *Sr33* gene [165].

Periyannan et al. [107] found that *Sr45* was effective against *Puccinia graminis* f. sp. *tritici* races prevalent in small populations in Australia and South Africa and the Ug99 race group, but the related detection was lower.

Kokhmetova and Atishova found that only the *Sr46* gene existed in the 338-K1-1//ANB/BUC/3/GS50A/4/422/5/BAYRAKTAR line when detecting 88 cultivars of spring soft wheat in Kazakhstan [166].

### 5.2. Application of Powdery Mildew Resistance Genes from Aegilops tauschii in Wheat Breeding

*Pm2a*, as a successfully cloned gene from *Aegilops tauschii*, is of great significance in wheat breeding for powdery mildew resistance. Švec et al. [167] identified the *Pm2* gene in 32 Polish wheat varieties. Agnieszka et al. [168] identified seven wheat varieties from Europe by establishing a multiplex PCR reaction and found that all wheat varieties contained the *Pm2* gene. Jimai 22 has been approved and widely promoted in Shandong Province and the northern part of Huanghuai; as of the summer harvest in 2020, the cumulative promotion area was 20 million hm^2^. Liangxing 66 has been promoted and planted in Shandong, central and southern Hebei, southern Shanxi, and Anyang, Henan, to the north of the Huanghuai winter wheat region [169]. Through genetic analysis and molecular marker detection, the above two varieties were found to carry the wheat powdery mildew resistance gene *Pm2* [170]. With the increase in the utilization frequency of *Pm2* in production, the frequency of the corresponding virulent species variation is also rising, resulting in an increasing risk of overcoming *Pm2* resistance.

With the acceleration of variety replacement, the varieties containing *Pm19* in actual production have gradually increased. Li et al. [171] identified 23 white powdery mildew-resistant materials and found that only one material contained *Pm19*. According to the identification results, *Pm19* was considered to have low resistance and should be used in combination with other resistance genes. Shi et al. [172] identified 61 reserve varieties of powdery mildew in China and found that 21 wheat varieties contained powdery mildew resistance genes, and four of them contained *Pm19*.

Although *Pm34* has not been cloned, it has been identified to contain this gene in wheat in actual production. Li et al. [173] identified 42 Yunnan wheat varieties using 20 wheat powdery mildew strains with different toxicity profiles and found that four varieties contained *Pm34*. Wang et al. [174] analyzed 305 wheat germplasm resources at home and abroad and found that 95 wheat varieties contained *Pm34*, accounting for 31.15%, including Lumai 5, Yanzhan 4110, Fengsheng 3, Jimai 22, CA9719, and Azulon.

El-Shamy et al. [175] used 12 Egyptian wheat varieties to identify the virulence of 52 powdery mildew strains and found that wheat varieties containing the *Pm35* gene had higher disease resistance. However, in actual production, there are few wheat varieties containing *Pm35*. Through toxicity monitoring and annual dynamic change analysis of wheat powdery mildew populations in Shaanxi Province, China, Liu et al. found that NCD3 wheat varieties containing *Pm35* had higher disease resistance [176]. Yan et al. identified 371 wheat materials from Hebei Province and found that only Pubing 01 contained the *Pm35* gene [177].

Due to the late discovery and cloning of *Pm58*, only the germplasm lines U6714-A-011 (Reg.No.GP-1023, PI682090) and U6714-B-056 (Reg.No.GP-1022, PI 682089) of the new powdery mildew resistance gene *Pm58* cultivated by Michigan State University using TA1662 and KS05HW14 are currently available, but the wheat yields of these two germplasm lines need to be improved [144].

### 5.3. Application of Other Disease-Resistant Genes from Aegilops tauschii in Wheat Breeding

Because the genetic research on wheat septoria tritici blotch resistance genes are relatively slow, *Stb5*, as the only localized gene of wheat septoria tritici blotch in *Aegilops tauschii*, has not been widely studied and utilized in production [178].

*Tsr3* is one of the four genes for wheat brown spot disease resistance officially mapped in *Aegilops tauschii* [179]. The research on *Tsr3* is less than that on wheat resistance to wheat septoria tritici blotch, and relevant production research reports have not been found yet.

## 6. Expectations

With the completion of the wheat gene map construction, breeders, on the one hand, have stepped up their research on the genes of common wheat itself; on the other hand, they have also excavated and utilized wheat-related plants. *Aegilops tauschii*, as a relative plant of wheat and an ancestor species of the wheat D genome, has a wider genetic diversity than that of the wheat D genome. Compared with the wheat A and B genomes, the D genome has the lowest degree of excavation. At present, the gene map of *Aegilops tauschii* has been basically constructed, and a reference-quality genome sequence for *Aegilops tauschii* has been available since 2017. It is of great significance to supplement wheat’s genetic resources and improve its genetic diversity by fully excavating and utilizing the disease resistance genes in it.

According to previous studies, we found that there is still a huge gap from the successful gene mapping or cloning of genes in *Aegilops tauschii* to the application of genes to actual production. The utilization of wheat disease resistance genes in *Aegilops tauschii* should be promoted from the following aspects: (1) Intensify the excavation of wheat disease resistance genes in *Aegilops tauschii*. Through the previous studies on stripe rust, leaf rust, and stem rust, it is found that there are still few disease resistance genes discovered in *Aegilops tauschii*. Understanding how to explore more disease resistance genes from *Aegilops tauschii* will still be a key topic for a long time in the future. We should innovate the research methods of gene discovery and accelerate the discovery of excellent resistance genes in *Aegilops tauschii* through association map analysis, map-based cloning, MutChromSeq, long reads, CRISPRs, and other modern methods. (2) Clone the discovered wheat resistance genes. It can be seen from the previous description that although many wheat disease resistance genes have been discovered and mapped in *Aegilops tauschii*, there are still few genes successfully cloned, and the disease resistance mechanism still needs to be further studied. Because the traditional map-based cloning technology is very time-consuming and entails a huge, laborious workload, and because it often takes many years to successfully clone genes, understanding how to clone disease-resistant genes quickly and efficiently is still a difficult challenge for the future. (3) Accelerate the application of cloned genes in wheat resistance breeding production. Previous studies have shown that only a few of the cloned disease resistance genes in *Aegilops tauschii* have been successfully applied to breeding production, while the most widely used disease resistance genes in actual production are still the first few genes cloned. With the large-scale application of single disease-resistant genes and the continuous emergence of new pathogenic races, many production varieties will rapidly lose disease resistance after several years of planting. In production, polygene polymerization breeding should be adopted to broaden the variety of disease resistance genes and reduce the loss of disease resistance genes as much as possible. Moreover, there may be genetic encumbrance among the resistance genes. Therefore, understanding how to successfully break this genetic encumbrance, speed up the transfer of cloned genes into the wheat genome, and cultivate new disease-resistant lines is still the top priority in the application of disease resistance genes from *Aegilops tauschii* in wheat breeding.

## Figures and Tables

**Figure 1 plants-12-00880-f001:**
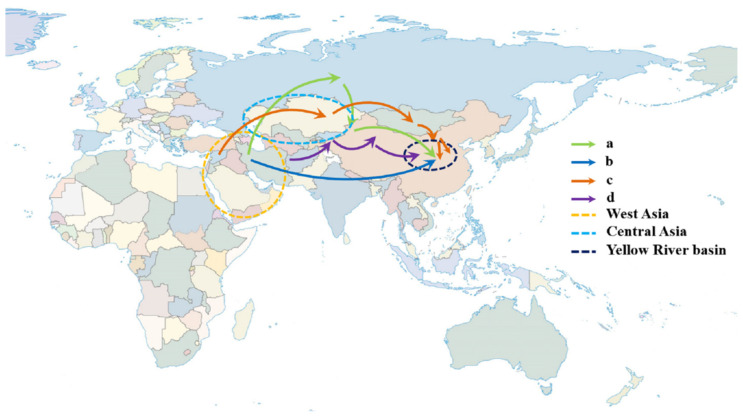
Road map of the introduction of *Aegilops tauschii* and wheat into China. The route of *Aegilops tauschii* introduced into China: (a) Middle East→Russia→Xinjiang, China→Yellow River Basin; (b) the ancient Silk Road in the Middle East; (c) northwest of West Asia→Eurasia→southern Siberia and Mongolia→Middle and Lower Yellow River Region; (d) Afghanistan or Central Asia Oasis →Northern Xinjiang→Yellow River Basin.

## Data Availability

Not applicable.

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
