# Peer review of "Advances in the Mining of Disease Resistance Genes from Aegilops tauschii and the Utilization in Wheat"

_plants, 2023, doi:10.3390/plants12040880_

Round 1

Reviewer 1 Report

Aegilops tauschii is the wild species ancestor of the D genome of hexaploid wheat (AABBDD). It is an important genetic resource for the mining and utilization of wheat disease resistance genes.  A review (177 references) on this problem is presented in manuscript.  

The article is well structured, it introduced the research progress of Aegilops tauschii, the discovery of   disease resistance genes (Lr, Yr, Sr, Pm, Stb, and Tsr), and their application in wheat breeding.

The article concludes with problems whose solution will contribute to the effective use of valuable resistance genes from   Aegilops tauschii for breeding.   

I believe that the publication of this review is appropriate.

Author Response

We appreciate your professional review of our paper. We sincerely thank you for your enthusiastic work. Have a good life!

Reviewer 2 Report

During the reading of presented paper, one big question was raised: Is the paper dedicated to the wheat breeding utilizing resistance genes identified/cloned in Aegilops tauschii or is it going just about genes in Aegilops tauschii genome? The title is pointing to the genes in Ae.tauschii, the abstract contains an aim dedicated to the wheat breeding, but the main text contains both. To be honest, even though it seemed to be clear after reading the abstract, after reading the whole text, I lost the main thought and I am now not sure, what was really going on. Was it just a time scale of resistance gene discoveries? Or does the review have a bigger purpose?

Authors cited 177 studies. On one hand, it is an evidence, that authors really tried to do their best and bring a complex view of this topic, on the other hand I would strongly recommend to reduce the list. Surprisingly, I miss some really important papers dedicated to Aegilops tauschii or other topics mentioned in the paper.

Writing a review focusing of (resistance) genes is always a challenge even in case of less complicated plant genomes and having a complex review and list of such genes can really facilitate future research. However, I also think, that presented review needs modifications. That's why I strongly recommend major revision to make the text clear for all readers. All my comments and recommendations are present is the attached file. 

Author Response

We appreciate your professional review of our paper. We sincerely thank you for your enthusiastic work and thank you again for your comments. Please see the attachment.

Reviewer 3 Report

line 35: ..are...?

line 271: ..., found to be...

line 364: ...contains resistance genes to....

line 403: ..contained...?

Author Response

(The authors gave the same response as above.)

Round 2

Reviewer 2 Report

Thank you for adding information I was asking for. 

Author Response

We appreciate your professional review of our paper. We sincerely thank you for your enthusiastic work and thank you again for your comments.